# Diamond Composites Produced from Fluorinated Mixtures of Micron-Sized and Nanodiamonds by Metal Infiltration

**DOI:** 10.3390/ma15144936

**Published:** 2022-07-15

**Authors:** Valery N. Khabashesku, Vladimir P. Filonenko, Rustem K. Bagramov, Igor P. Zibrov

**Affiliations:** 1Department of Materials Science and Nanoengineering, Rice University, Houston, TX 77005, USA; 2Vereshchagin Institute of High Pressure Physics RAS, 142190 Moscow, Russia; bagramov@mail.ru (R.K.B.); zibrov@hppi.troitsk.ru (I.P.Z.)

**Keywords:** fluorinated diamond powders, high-pressure–high-temperature sintering, polycrystalline diamond composites

## Abstract

Improving the operating performance of superhard composites is an important and urgent task, due to a continuing industrial need. In this work, diamond composites with high wear resistance were obtained by sintering fluorinated mixtures of micron-sized diamonds with nanodiamonds at high pressures and temperatures (7–8 GPa, 1550–1700 °C). Aluminum and cobalt powders were added to the diamond mixture to activate the process. The external infiltration of nickel into the diamond layer was carried out additionally during the sintering process, and the effects of nickel infiltration on the structure and properties of composites were studied. The metal melt ensured the mass transfer of carbon within a volume, and the formation of a strong diamond framework. The composition of the additives was selected in such a way that the binding phase became ultimately composed of the intermetallic AlNi_x_Co_1−x_(x ≤ 1). The Young’s modulus of composites synthesized in this way had a value of 850 GPa, and their wear resistance when turning white granite was more than twice as high as that of premium commercial PDC. The obtained results thus demonstrate that by using nickel to increase melt infiltration into diamond-based composites, the mechanical properties of Al/Co/fluorinated diamond compositions, studied previously, can be further improved.

## 1. Introduction

Diamond composites are superhard materials consisting of polycrystalline diamond matrix and reactive additives that promote sintering and/or provide improved mechanical characteristics. Two basic synthetic approaches to diamond composites are in use: (i) high-pressure–high-temperature (HPHT) sintering of powder mixtures of diamond and additives, and (ii) external infiltration of additives from the substrate layer into the diamond layer.

Carbide-forming elements, such as silicon and titanium, can be used as sintering promoters [1,2,3,4]. They are capable of forming refractory carbide layers around diamond particles. Such composites demonstrate high heat-resistance, but are too brittle. For comparison, metals (Co, Ni, and others) that do not form carbides can dissolve carbon, so that their melts facilitate the mass transfer and recrystallization of carbon, to result in the formation of a strong diamond carcass under high pressure and temperature [5,6].

Two-layer polycrystalline diamond composites (PDC) are most often used for drilling and cutting tools [7,8]. The top diamond layer of PDC consists of metal infiltrated diamond material, while the substrate layer is made of a hard alloy (for example, WC-8Co). During the high-pressure synthesis, liquid cobalt from the hard alloy layer penetrates into the diamond layer. The main disadvantage of such composites is their low thermal stability. This is due to the large difference in the thermal expansion coefficients of the diamond and metal phases in the PDC, which causes critical local thermomechanical stresses during operation. In addition, at operating temperatures above 700 °C, the cobalt phase initiates the reverse transition of diamond into graphite [9,10,11,12], which also reduces the mechanical properties and operating performance of the tool.

Fluorination of the initial diamond mixtures and the use of aluminum in addition to cobalt provided a new approach for HPHT sintering of PDC. The thermal stability of such PDCs has been improved due to the formation of binder phases of the AlCo intermetallic compound and AlCo_3_C ternary carbide [13]. The wear resistance of a PDC diamond layer containing such binder phases was found to be several times higher than that of commercial premium composites [14].

It was shown [15] that at pressure of 8 GPa and temperature of 1800 °C, the nickel melt wetted the surface of graphite and diamond better than the cobalt melt, so the depth of nickel penetration into the microdiamond layer was two times higher than that of cobalt. Accordingly, nickel-bonded diamond composites showed higher strength; however, their wear resistance was inferior to cobalt-bonded composites during the sandstone turning.

Facilitating the infiltration of metal into diamond powder feedstock presents a very important task for the development of diamond–metal composites. In this work, we assessed the prospects of using nickel to increase melt infiltration into diamond-based composites, so that the Al/Co/fluorinated diamond compositions, studied previously, can be further improved.

After the introduction of nickel into fluorinated mixtures of micro- and nanodiamonds with additives of aluminum, and cobalt with aluminum, and their treatment under high pressure and temperature, a comparative study of the structure and mechanical properties of synthesized PDCs was carried out.

## 2. Materials and Methods

Synthetic diamond powder of 8–12 μm and diamond nanopowder of ~90 nm primary particle size (obtained by mechanical grinding) were purchased from NanoDiamond Products (NDP, Shannon, Ireland). The powders were mixed in a ratio of 7:3, annealed in hydrogen at 600 °C, and then fluorinated with a gas mixture of 10% F_2_/90% He at a temperature of 340 °C [16]. Powders of aluminum (0.5–2.0 μm) and cobalt (1–3 μm) were added to this initial mixture of fluorinated diamonds (FDM). The FDM mixture with 3 wt.% aluminum was designated as FDM-3Al, and the FDM mixture with 2 wt.% aluminum and 6 wt.% cobalt was designated FDM-2Al-6Co. The process of preparation and analysis of such mixtures is described in more detail in [14].

For sintering experiments at high pressures, equipment of the “Toroid” type, developed at Vereshchagin Institute of High Pressure Physics RAS, Moscow, Russia, was used [17]. The “Toroid” chamber and the scheme of the reaction cell are shown in Figure 1. The sintering pressure was 7–8 GPa, the temperature was 1550–1700 °C, and the holding time was 30 s. The synthesized disk-shaped samples had the dimensions of 5 mm diameter and 3 mm height.

Scanning electron microscopy (JEOL JSM-6390 from JEOL USA Inc., Peabody, MA, USA) was used to study the microstructure of the samples and the elemental composition of the metal binder in the composite. X-ray phase analysis was performed using a HUBER diffractometer (From HUBER Diffraktionstechnik GmbH, Rimsting, Germany). The elastic moduli were calculated from the propagation velocities of the longitudinal and transverse ultrasonic waves. The wear resistance of experimental samples was compared with commercial US Synthetic PDC composites by turning WC-8Co hard alloy and white granite.

## 3. Results and Discussion

Superhard composites with a high level of mechanical properties can be obtained by sintering diamond powders in the pressure and temperature range of thermodynamic stability of diamond. However, the implementation of this strategy is complicated by the possibility of the diamond for undergoing a reverse phase transition to graphite (graphitization), occurring at high temperatures, in the pores between diamond particles. Although the pressure in the points of contact of diamond grains can exceed 100 GPa [15], it is significantly lower in the pores between diamond particles than in the surrounding media of the high-pressure cell.

The graphite phase was found in FDM samples that were sintered without the addition of activators at 7–8 GPa and 1500 °C. The extent of graphite phase formation depended both on temperature and time of exposure to this temperature. To suppress the graphitization, the addition of aluminum to diamond powders can be used. During the sintering process, aluminum melt fills the pores between the diamond crystals and creates the Al-C-F fluid, which promotes carbon transfer and recrystallization of the diamond phase and also prevents a reverse phase transition. It has been demonstrated [18] that the addition of aluminum facilitates the growth of fluorinated 10 nm nanodiamonds to micron-sized crystals. At the same time, aluminum does not cause significant enlargement of 90 nm fluorinated nanodiamonds; however, it promotes formation of good inter-grain boundaries in FDM.

Figure 2a shows a characteristic diffraction pattern of a sample obtained by sintering a mixture of FDM-3Al. Transcrystalline fracture of micron particles (Figure 2b) indicates their strong interfacial contact with nanodiamonds, the size of which remains smaller than 0.1 μm.

The presence of graphite in the samples sintered from the FDM-3Al mixture may be due to the low content of fluorine in the initial mixture, which is proportional to the total surface of all diamond particles. The fluorine content for a 90 nm fluorinated diamond is 3.5 at.% [19], while for the FDM mixture its content is approximately 1 at.%. Thus, the amount of the produced Al-C-F fluid is not sufficient for complete suppression of the graphitization. In addition, the excess aluminum forms carbide of Al_4_C_3_ stoichiometry, which is undesirable, since its reaction with water will lead to deterioration of the properties of the composite.

The addition of cobalt helps to eliminate the disadvantages mentioned above. Composites obtained from homogeneous mixtures of FDM plus aluminum and cobalt have increased heat resistance and wear resistance, comparable to commercial premium composites [20]. It has also been shown that the FDM-3Al mixture layer can be sintered with a hard alloy WC-Co substrate. In this case, cobalt penetrates into the FDM-3Al layer and increases its wear resistance to a level significantly higher than that of the corresponding homogeneous mixture [14]. The cobalt melt during infiltration promotes the mutual rearrangement of diamond grains, promotes their dense packing, and promotes the formation of strong interfacial contacts between them. The degree of infiltration depends on the permeability of the diamond layer and on thermobaric conditions. The depth of cobalt penetration into the diamond layer may be limited by the formation of graphite in the pores between the diamond particles.

It has been shown [15] that at sintering conditions of 8 GPa and 1800 °C the penetration depth of nickel into the layer of micron-sized diamonds after 9 s of exposure was two times larger than that of cobalt. To infiltrate nickel into diamond layers in this work, a nickel foil was placed onto the surface of a 3 mm tablet. Infiltration was carried out into layers of fluorinated microdiamond, FDM, and FDM-3Al mixture at 8 GPa and 1600 °C. The heating rate was 50 and 10 °C/sec. It has been established that the infiltration depth decreases in the following sequence of diamond layers: micron diamond < FDM < FDM-3Al. Obviously, this is due to the presence of small diamond particles and metal in the pores between micron-sized diamond particles, which become narrower and less permeable. With a decrease in the heating rate, the infiltration depth also decreases. For example, at a heating rate of 50 °C/sec, nickel penetrated the tablets of fluorinated micro-diamonds (without additives) to a depth of 2.5–3.0 mm. Additionally, the depth of infiltration into the mixture of FDM-3Al at a heating rate of 10 °C/sec was found to be smaller: 0.9–1.2 mm.

Nickel infiltration to the full depth is necessary to calculate elastic modules from the propagation velocities of ultrasonic waves through the sample; therefore, nickel foil was placed on both end surfaces of diamond layers. Infiltration experiments, conducted at 8 GPa, 1600 °C, and heating rates of 50 °C/sec, showed that nickel penetrated the diamond layers consisting of FDM, FDM-3Al, and FDM-2Al-6Co compositions to the full depth. Figure 3 shows the fracture surface of an FDM sample infiltrated with nickel. It can be seen that the destruction of diamond grains is transcrystalline. The nickel content in the volume of the layer was found to be approximately 10% by weight. The amount of nickel that penetrated into FDM-3Al and FDM-2Al-6Co layers was significantly lower (Figure 4a), because during the heating process the metal powders of Al and Co present in these mixtures also melted and filled the pores between micron particles.

The metal concentration profiles for the FDM-2Al-6Co sample infiltrated with Ni at 8 GPa and 1600 °C are shown in Figure 4a. In this sample, some amounts of aluminum and cobalt are pushed from the surface layer to the central part. The nickel concentration varies more significantly, decreasing from 3 wt.% in the surface layer to approximately 0.3 wt.% in the middle of the sample. After infiltration of nickel, a strong diamond frame is formed and the resulting composites demonstrate a transcrystalline type of fracture. That is, the destruction propagates through the micron-sized diamond crystals, and not along their interfacial boundaries. The total amount of cobalt and nickel measured along the height of the sample varies insignificantly with respect to aluminum, and thus, the elemental composition of the metallic phase remains in the region of homogeneity of the AlNi_x_Co_1−x_(x ≤ 1) intermetallic compound. The formation of ternary carbides in such samples has not been established.

X-ray diffraction analysis of the samples obtained by infiltration of nickel into the FDM-3Al layer showed that the binder phase in such composites consists of the AlNi intermetallic compound, which has a wide homogeneity region on the phase diagram. The broadening of the XRD peaks of this intermetallic compound (Figure 4b) can be associated with its large deviations from the stoichiometric composition while maintaining the structural type of the crystal lattice. Apparently, the homogenization of the elemental composition of the intermetallic compound in the whole volume of the sample cannot be completed during the limited time of sintering.

Figure 5 presents SEM images of a composite obtained by infiltration of nickel into a mixture of FDM-2Al-6Co at 8 GPa and 1600 °C (heating rate 50 °C/sec). Nanodiamonds filling the pores between microdiamonds grow up to 0.2–0.3 μm in size during sintering. Nanocrystals sandwiched between the faces of micron particles completely recrystallize and promote the formation of diamond–diamond bonds. Figure 5c shows that in the regions of the triple junction of micron-sized diamond grains, intermetallic inclusions have dimensions of the order of 1 μm. At the same time, nanosized inclusions of the AlNi_x_Co_1−x_(x ≤ 1) intermetallic compound decorate well-formed boundaries between micron-sized diamond grains.

Table 1 demonstrates the results of measurements of the elastic moduli of samples obtained from FDM-2Al-6Co by sintering, and composites produced from this mixture with additional infiltration of nickel during sintering at 7.5 GPa and 1550 °C. Young’s modulus of the samples 1 and 2 prepared for comparison by nickel infiltration is slightly higher than for the samples obtained by sintering, although these composites contain a larger amount of binding phase.

It can be seen that the samples demonstrate some differences in the propagation velocities of longitudinal and transverse waves; however, Young’s moduli in all composites are at a fairly high level of ~850 GPa. Since the high speed of sound is provided by the quality of the boundaries of the diamond framework, this means that diamond–diamond boundaries are well formed and the influence of the intergranular metal phase is accordingly small. In addition, the diamond framework also determines the level of wear resistance of composites.

Wear resistance tests were carried out by turning cylindrical blanks of WC-8Co hard alloy and white granite. Figure 6 and Figure 7 show the wear spots of the tested samples sintered from the FDM-2Al-6Co mixture without nickel infiltration and with nickel infiltration. It can be seen that after turning the hard alloy (depth of cut 0.5 mm, test time 10 min) the experimental FDM-2Al-6Co sample, obtained with nickel infiltration, shows a smaller wear spot than the commercial PDC. When turning white granite (depth of cut 1.0 mm, test time 45 min), the difference in work performance of tested samples becomes even more evident.

When turning white granite, the wear of experimental samples made by sintering the FDM-2Al-6Co mixture was found to be more than two times higher than that of commercial PDC. With the additional infiltration of nickel into the FDM-2Al-6Co diamond layer, the elastic modules of the composites did not change, and the resistance to wear increased. For the composite made of FDM-2Al-6Co (Figure 7b) without nickel infiltration, the wear spot shows chips in the cutting zone. This is typical for materials with increased fragility. As shown in Figure 7c, additional infiltration of nickel eliminates this disadvantage and leads to an increase in the wear resistance of composites. This is because infiltration, proceeding by propagation of nickel melt into the diamond layer, facilitates the reassembling of diamond particles and more optimized packing of neighboring diamond crystals in the composite, which makes it difficult for microcracks to spread.

## 4. Conclusions

Diamond composites, based on the fluorinated mixtures of micron-sized and nanodiamonds as a feedstock, were synthesized under conditions of high pressures and temperatures (7–8 GPa, 1550–1700 °C) with the admixture of powdery sintering aids (aluminum, cobalt) and additional external infiltration of nickel during the sintering process.

After infiltration of nickel into a diamond layer comprised of fluorinated diamonds and aluminum, an AlNi binding phase is formed, while adding the aluminum and cobalt to fluorinated diamonds results in formation of an intermetallic compound AlNi_x_Co_1−x_(x ≤ 1).

Young’s modulus of the samples sintered from the FDM-2Al-6Co mixture without nickel infiltration was found to be at a fairly high level of ~850 GPa. Wear resistance of these composites when turning white granite was more than two times higher than that of commercial PDC. After sintering these mixture compositions along with external nickel infiltration, Young’s modulus of produced composites did not decrease and their wear resistance during turning of hard alloy and white granite increased due to reduced brittleness and lack of microchips.

The obtained results suggest that the method of external infiltration of nickel into a diamond layer composed of fluorinated micron-sized and nanodiamonds, with the addition of aluminum and cobalt, can be recommended for application in the production of tool-grade diamond composites.

## Figures and Tables

**Figure 1 materials-15-04936-f001:**
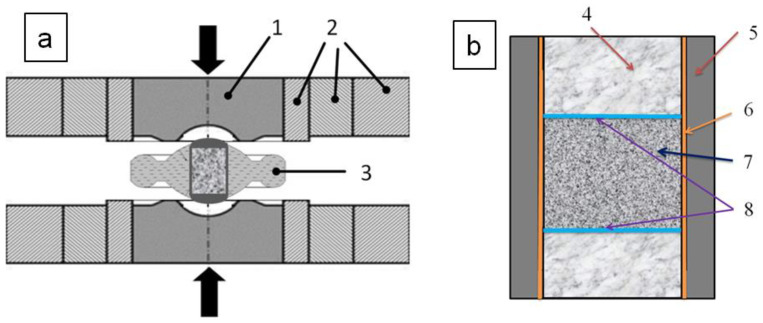
“Toroid” type high-pressure chamber. (**a**) Chamber assembly scheme: 1—hard alloy punch, 2—steel rings, and 3—limestone container; (**b**) High pressure cell: 4—a tablet made from mixture of graphite and boron nitride, 5—graphite heater, 6—Ta-foil, 7—a mixture of diamond powders, and 8—Ni-foil.

**Figure 2 materials-15-04936-f002:**
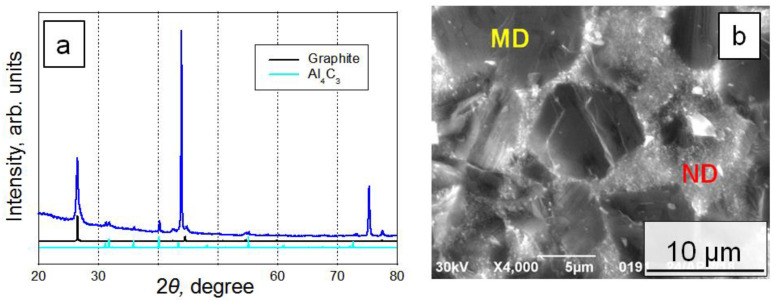
(**a**) X-ray diffraction pattern and (**b**) SEM image of the sample obtained by the sintering of the FDM-3Al at 8 GPa and 1700 °C (MD—micron-size diamond, ND—nanodiamond).

**Figure 3 materials-15-04936-f003:**
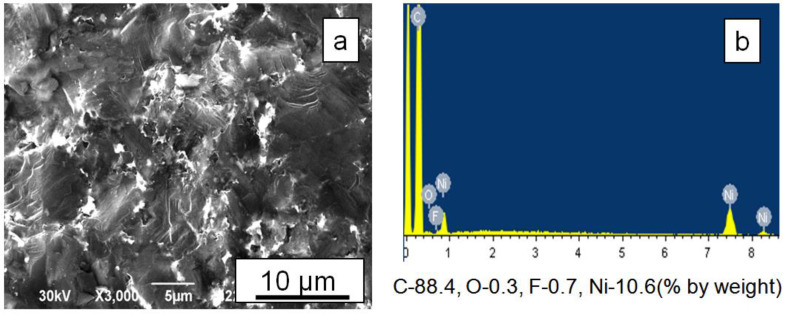
(**a**) SEM image and (**b**) elemental analysis of the fracture surface of the FDM sample infiltrated with nickel at 7 GPa and 1600 °C.

**Figure 4 materials-15-04936-f004:**
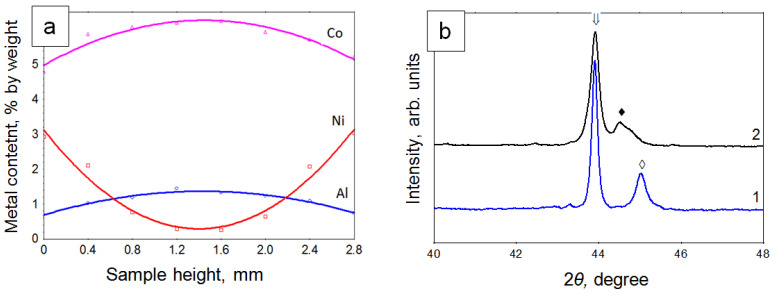
(**a**) Concentration profiles for the FDM-2Al-6Co sample infiltrated with Ni (⇓—111 diamond, ♦—NiAl, ◊—CoAl); (**b**) X-ray diffraction patterns of composites obtained at 8 GPa, 1600 °C: (1) FDM-3Al-9Co homogenous mixture and (2) FDM-3Al sample infiltrated with Ni.

**Figure 5 materials-15-04936-f005:**
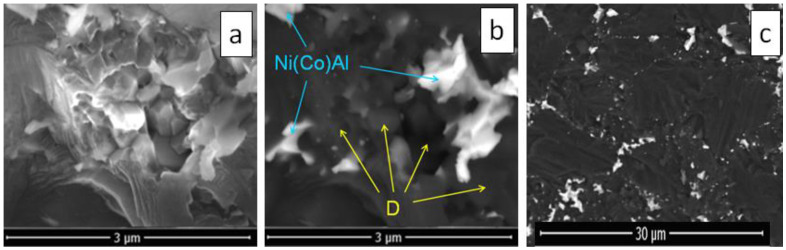
SEM images of a composite obtained by infiltration of nickel into a mixture of FDM-2Al-6Co at 8 GPa and 1600 °C. Submicron diamonds and AlNi_x_Co_1−x_ intermetallic compound located between micron diamond particles: (**a**) image in reflected electrons; (**b**) the same image in phase contrast mode; and (**c**) fracture surface in phase contrast mode.

**Figure 6 materials-15-04936-f006:**
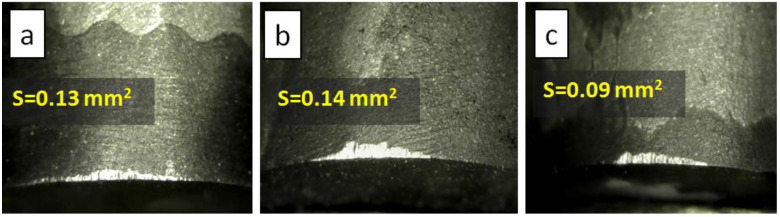
Photographs of wear spots of diamond composites taken after turning of hard alloy WC-8Co blanks: (**a**) commercial PDC from US Synthetic; (**b**) composite obtained from a mixture of FDM-2Al-6Co, and (**c**) nickel infiltrated FDM-2Al-6Co composite.

**Figure 7 materials-15-04936-f007:**
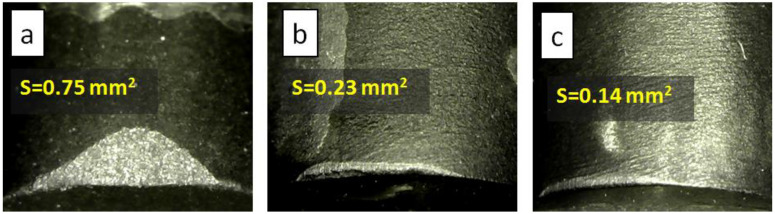
Photographs of wear spots of diamond composites taken after turning of white granite: (**a**) commercial PDC from US Synthetic; (**b**) composite obtained from a mixture of FDM-2Al-6Co; (**c**) nickel infiltrated FDM-2Al-6Co composite.

**Table 1 materials-15-04936-t001:** Elastic moduli of composites obtained from FDM-2Al-6Co and nickel at 7.5 GPa and 1550 °C.

Nickel Addition	Longitudinal Velocity V_l_, km/s	Transverse Velocity V_t_, km/s	Shear Modulus G, GPa	Bulk Modulus B, GPa	Young’s Modulus E, GPa
Without nickel	16.21	9.93	350	467	842
Infiltration of nickel into sample 1	16.87	9.32	335	650	854
Infiltration of nickel into sample 2	15.88	9.77	358	469	863

## Data Availability

Not applicable.

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
