# Peer review of "Diamond Composites Produced from Fluorinated Mixtures of Micron-Sized and Nanodiamonds by Metal Infiltration"

_materials, 2022, doi:10.3390/ma15144936_

Round 1

Reviewer 2 Report

The paper seeks to introduce an approach ‘’ Diamond composites produced from fluorinated mixtures of micron-sized and nanodiamonds by metal infiltration’’ However, the authors should consider to improve upon the quality to further highlight and emphasis. 

1.    Based on the comprehension of what constitutes an abstract, consider adding one or two lines introducing the problem you are trying to solve.

2.    Again, add one or two lines highlighting the importance of your study at the end of the abstract.

3.    The maximum numbers of words allowed in the keywords section is three. Consider reducing the ones with more than three words to three maximum.

4.    The introduction needs to be improved by relating to the mechanics of the studied materials and their mechanical characteristics. The references to be included are: 10.1016/j.compstruct.2021.114698 and 10.1016/j.jiec.2022.06.023.

5.     State the aim of the study clearly at the end of the introduction. It is supposed to summarize the description of what is being done.

6.    What were the accelerating voltage used in the SEM analysis, the working range, and the scale bar used?

7.    One standard of writing style should be adopted, uniformity is vital in scientific writing. If you consider writing fig., it should run through and vice versa but not alternating between fig. and figure.

8.    The SEM image in figure 2 and 3 have two magnification footers, the one automatically generated (5 µm) and the one you manually created (10 µm). Why are there differences in the two magnifications?

9.    You stated in the paragraph under figure 2 that “The presence of graphite in the samples sintered from the FDM-3Al mixture may be due to the low content of fluorine in the initial mixture, which is proportional to the total surface of all diamond particles”. If it may, then it could as well might not. Do you have any reference to that effect since you are not so sure?

10. Remove the word “at” in that same paragraph described in point 8 that “while for the FDM mixture its content is about 1 at.%”.

11. Reference the statement “The addition of cobalt helps to eliminate the disadvantages mentioned above”.

12. Put a space between each variable and its corresponding unit.
